# Cryptoasset networks: Flows and regular players in Bitcoin and XRP

**Hideaki Aoyama**[1,2,3ℴ], **Yoshi Fujiwara**[4ℴ]*, **Yoshimasa Hidaka**[2,5,6ℴ], **Yuichi Ikeda**[1ℴ]

**1** Graduate School of Advanced Integrated Studies in Human Survivability, Kyoto University, Kyoto, Japan, **2** RIKEN iTHEMS, Wako, Japan, **3** Research Institute of Economy, Trade and Industry, Tokyo, Japan, **4** Graduate School of Information Science, University of Hyogo, Kobe, Japan, **5** Institute of Particle and Nuclear Studies, KEK, Tsukuba, Japan, **6** Graduate University for Advanced Studies (Sokendai), Tsukuba, Japan

ℴ These authors contributed equally to this work.
\* yoshi.fujiwara@gmail.com

**Data Availability Statement:** All relevant data are within the paper and its Supporting information file.

**Funding:** Y.I.: A grant "University Blockchain Research Initiative" provided by Ripple, Inc. to Kyoto University Y.F.: JSPS KAKENHI Grant

## Abstract

Cryptoassets flow among players as recorded in the ledger of blockchain for all the transactions, comprising a network of players as nodes and flows as edges. The last decade, on the other hand, has witnessed repeating bubbles and crashes of the price of cryptoassets in exchange markets with fiat currencies and other cryptos. We study the relationship between these two important aspects of dynamics, one in the bubble/crash of price and the other in the daily network of crypto, by investigating Bitcoin and XRP. We focus on "regular players" who frequently appear on a weekly basis during a period of time including bubble/crash, and quantify each player's role with respect to outgoing and incoming flows by defining flow-weighted frequency. During the most significant period of one-year starting from the winter of 2017, we discovered the structure of three groups of players in the diagram of flow-weighted frequency, which is common to Bitcoin and XRP in spite of the different nature of the two cryptos. By examining the identity and business activity of some regular players in the case of Bitcoin, we can observe different roles of them, namely the players balancing surplus and deficit of cryptoassets (Bal-branch), those accumulating the cryptoassets (In-branch), and those reducing it (Out-branch). Using this information, we found that the regime switching among Bal-, In-, Out-branches was presumably brought about by the regular players who are not necessarily dominant and stable in the case of Bitcoin, while such players are simply absent in the case of XRP. We further discuss how one can understand the temporal transitions among the three branches.

## Introduction

In recent years, decentralized and open information systems based on blockchain technology have attracted much attention in financial applications. Crypto assets are recorded and managed by blockchain technology. Since the blockchain is a mechanism that makes it extremely difficult to alter data, there is almost impossible to lose crypto assets due to system failure or hacking. Therefore, in recent years, crypto assets have been increasingly held for investment

Numbers, 19K22032 and 20H02391, and the Nomura Foundation (Grants for Social Science) All the funders had no role in study design, data collection and analysis, decision to publish, or preparation of the manuscript.

**Competing interests:** The authors have declared that no competing interests exist.

purposes, with the expectation that the price of crypto assets will rise. In addition to investment purposes, crypto assets are becoming increasingly popular as a means of payment and remittance, and a great deal of transaction data has been accumulated. On the other hand, many crypto assets are exchanged in markets with fiat currencies. Today we observe that the market capitalization is so huge, and the price is highly volatile, having a considerable impact on global asset allocation. It is interesting to clarify the reality of the financial transaction networks of crypto assets such as Bitcoin and XRP from network science. Crypto assets such as Bitcoin and XRP comprise a complex network with nodes being the users or the node IDs and directed edges being transactions. The network is giant with billions and trillions of nodes and edges and temporally drastically changing.

The relationship between network characteristics and prices has been the subject of a relatively large number of preceding studies for Bitcoin and Ethereum; see the review paper for an overview of preceding studies for Bitcoin and Ethereum [1]. The authors reviewed preceding studies in terms of three aspects: network modeling, network profiling, and network-based detection. Preceding studies on Bitcoin and Ethereum considered most relevant to our paper are outlined below. Kondor et al. reconstructed the Bitcoin transaction network between users and analyzed changes of essential features of the time variation of the network [2]. The authors showed how structural changes in the network accompany significant changes in the Bitcoin price by applying principal component analysis to the matrix constructed from the daily snapshots of the transaction network. There are also many studies from such a viewpoint of complex networks on Bitcoin. See [3–17] for example, and references therein. Akcora et al. introduced a novel concept of chainlets as features to predict Bitcoin price [18]. The authors studied the role of chainlets on Bitcoin price formation and dynamics. They identified specific types of chainlets that exhibit the most decisive influence on Bitcoin price. Griffin et al. studied whether Tether, known as one of the stable coins pegged to the U.S. dollar, influenced Bitcoin price during the 2017 boom [19]. They found that purchases with Tether are timed following market downturns and result in sizable increases in Bitcoin prices. Recent review and work [20] studies cryptocurrency market in a wider perspective to find multiscale characteristics of such emerging global market of exchanges.

We note, however, that there is little prior research on XRP. Moreno-Sanchez et al. proposed an algorithm to group wallets based on actual data on the Ripple network graph [21]. The authors deanonymized the operators of the observed wallets clusters and reconstructed the financial activities of deanonymized Ripple wallets. After this, the same authors studied the structure and evolution of the Ripple network and investigated its vulnerability to devilry attacks that affect the IOU credit of linnet users' wallets [22].

Our paper explores the relationship between transaction networks and prices based on three studies. Fujiwara et al. studied how Bitcoin flows among users to understand the structure and dynamics of the crypto asset at a global scale [23]. They compiled all the blockchain data of Bitcoin from its genesis to the year 2020, identified users from anonymous addresses of wallets, and constructed monthly snapshots of networks by focusing on regular users as big players. They conducted the bow-tie structure analysis and Hodge decomposition to locate the users in the entire crypto flow's upstream, downstream, and core. Additionally, they revealed principal components hidden in the flow by using non-negative matrix factorization. Moreover, they found that the bow-tie structure and the principal components were relatively stable among those big players. Ikeda revealed the reality of the financial transaction network of XRP by studying the correlation between network characteristics and price [24]. They built monthly XRP transaction networks from January 2013 to September 2019. To reveal the essential characteristics of the transaction network, they calculated various network centralities. After identifying the essential characteristics of XRP, they studied network motifs. Network motifs are

small topological patterns such as triangular sub-graphs that recur in a network significantly more often than expected by chance. These motifs were the more prevalent during the bubble-forming periods of 2014 and 2018 and less prevalent throughout the rest of the year. The statistically significant triangular motifs were identified by comparing the observed ratio with the theoretical expectation. Aoyama et al. proposed a new index: the "Flow Index" motivated by the behavior of some nodes with histories of large transactions [25]. The Flow Index is a pair of indices suitable for characterizing transaction frequencies as a source and destination of a node. Using this Flow Index, they studied the global structure of the XRP network and constructed a bow-tie/walnut structure. These studies provided a solid basis for further investigating the temporal change of asset flow, entry and exit of big players, etc.

It is essential to understand how crypto assets flow in the network, in particular, flow among "regular players", and how the structure and dynamics of the network are related to the exchange markets' volatile behavior. By regular players, we mean the users who appear frequently in transactions on a regular basis during a given period. In the case of Bitcoin, a player is a user who possesses one or more wallets, while in the case of XRP, a player is actually a node ID which transacts XRP frequently and regularly. We shall give our precise definition of regular players. For our study, we propose an index called "flow-weighted frequency" to identify regular players based on their activity in terms of frequency and amount of transactions with both of incoming and outgoing ones. We study the temporal structure of transactions made by those regular players and analyze the network structure of the money flow made among the active players. We focus on singular market behavior. Primarily, both Bitcoin and XRP had a price hike in January 2018. Our most interesting question is to characterize the significant price changes by the frequency and amount of transactions. The problem we are particularly interested in is the relationship between the temporal structure of crypto assets transactions and the significant changes in prices. Therefore, our goal is to explain the significant price change in January 2018 using the proposed "flow-weighted frequency" index.

This paper organizes as follows. In *Materials and methods*, we first explain data for Bitcoin and XRP. Then we explain the *Definition of Flow-weighted Frequency*. In *Results*, we show various findings of the three-branch structure for transactions of Bitcoin and XRP. We explore Users/Addresses in the three branches. Based on these results, we reveal the correlations between the characteristics of the transactions captured by the Flow-weighted index and the price of crypto assets. In *Discussion*, Interpretation of the three-branch structure and the relation to price data are discussed. Finally, we conclude this study in *Conclusion*.

## Materials and methods

### Data: Cryptoasset of Bitcoin

We use the entire data set of the Bitcoin blockchain from the genesis block (the first block issued on January 9, 2009) until the block of height 693,999 (issued on August 3, 2021). Each block contains a number of transactions. Each transaction is a transfer of a certain amount of BTC (monetary unit of Bitcoin) from one or more addresses to others. An address is a kind of wallet possessed by a user, who can be an individual or, more often, an agent doing business activities such as Exchanges, Services, Gambling, mining and so forth. Mining is also called a proof-of-work which accepts a new block to the blockchain by a consensus algorithm in a peer-to-peer network in the decentralized system of Bitcoin. A miner can be interpreted as an issuer of Bitcoin.

A user can and quite often possesses multiple of addresses. In principle, it is not possible to identify users from addresses. However, if addresses appear as input in a transaction, one can immediately conclude that those different addresses belong to the same user. By examining the

entire history of all the transactions up to a certain point of time, one can construct a list of correspondence from addresses to users at that point of time. This simple but useful method to identify users from addresses was proposed by [3] and has been extensively used in the literature (see [2, 6], and the data of [26, 27], for example). See also [23, Sec.2.1 and Appendix A]. By "players" we mean users in the case of Bitcoin.

Each transaction has the following data items:

- Timestamp (defined by the time of mining for the block that includes the transaction) given in UTC (Coordinated Universal Time)

- Source, i.e., a user who sends a certain amount of Bitcoin

- Destination, i.e., a user who receives the amount of Bitcoin

- Amount, i.e., the amount of transaction

The timestamp is the record for the block, not for the transaction itself, so it includes uncertainty within a few hours at most. This is not a problem for our study, as we shall see shortly.

We did not include the mining data contained in the blockchain in order to focus on the flow of Bitcoin among users. Additionally, we discard the changes, i.e., money returned to the sender as the balance of the amount sent, to exclude the self-loops, i.e., the transactions with identical source and destination, which are not of our interest in this study.

## Data: Cryptoasset of XRP

Data we have used is based on Ledger data, covering 2013–01–02 to which is a list of direct transactions. For each transaction, various values of properties ("items") are given. Our program to extract these Ledger data yields different number of items depending on the year. The largest number of items is 964 for 2016 and the smallest is 55 for 2019, and there are 367 items for 2017 and 292 items for 2018. The union of the items of all the years contains 1,354 items, whose breakdown is: 68 items of "destination_balance_changes[x]" 654 items of "source_balance_changes[x]" and 16 other items, which we call "core" items. The core items are the following: "amount", "currency", "delivered_amount" "destination" "destination_tag" "executed_time" "invoice_id" "issuer" "ledger_index" "max_amount" "source" "source_currency" "source_tag" "transaction_cost" "tx_hash" "tx_index".

The intersection of all the Ledger data is made of these core items and "destination_balance_changes[x].counterparty", "destination_balance_changes[x].currency", "destination_balance_changes[x].value" whose $x = 0, 1, \cdots 9$, and "source_balance_changes[x].counterparty", "source_balance_changes[x].currency", "source_balance_changes[x].value" whose $x = 0, 1$, totaling 55 items.

In selecting the XRP direct transactions from this data, we apply the following filters:

1. Items "*.amount" has to be either "XRP" or empty.

2. Items "*.amount" and "*.value" that are not empty have the same value of XRP.

After these filterings, we have the data of transactions, each of which has the following items:

- Executed time given in UTC

- Source, i.e., node who sends a certain amount of XRP

- Destination, i.e., node who receives the amount of XRP

- Amount, i.e., the amount of transaction

Thus the transaction data for XRP are quite similar to the one for Bitcoin. By "players" we mean nodes in the case of XRP.

## Method: Target period, active and regular players

Both Bitcoin and XRP prices are known to have peaked in December 2017 to January 2018, after which the bubble crashed. The price data of Bitcoin and XRP are shown in Fig 1, which is obtained using the Poloniex API [28]. The plots show the sharp peak on December 17 2017 for

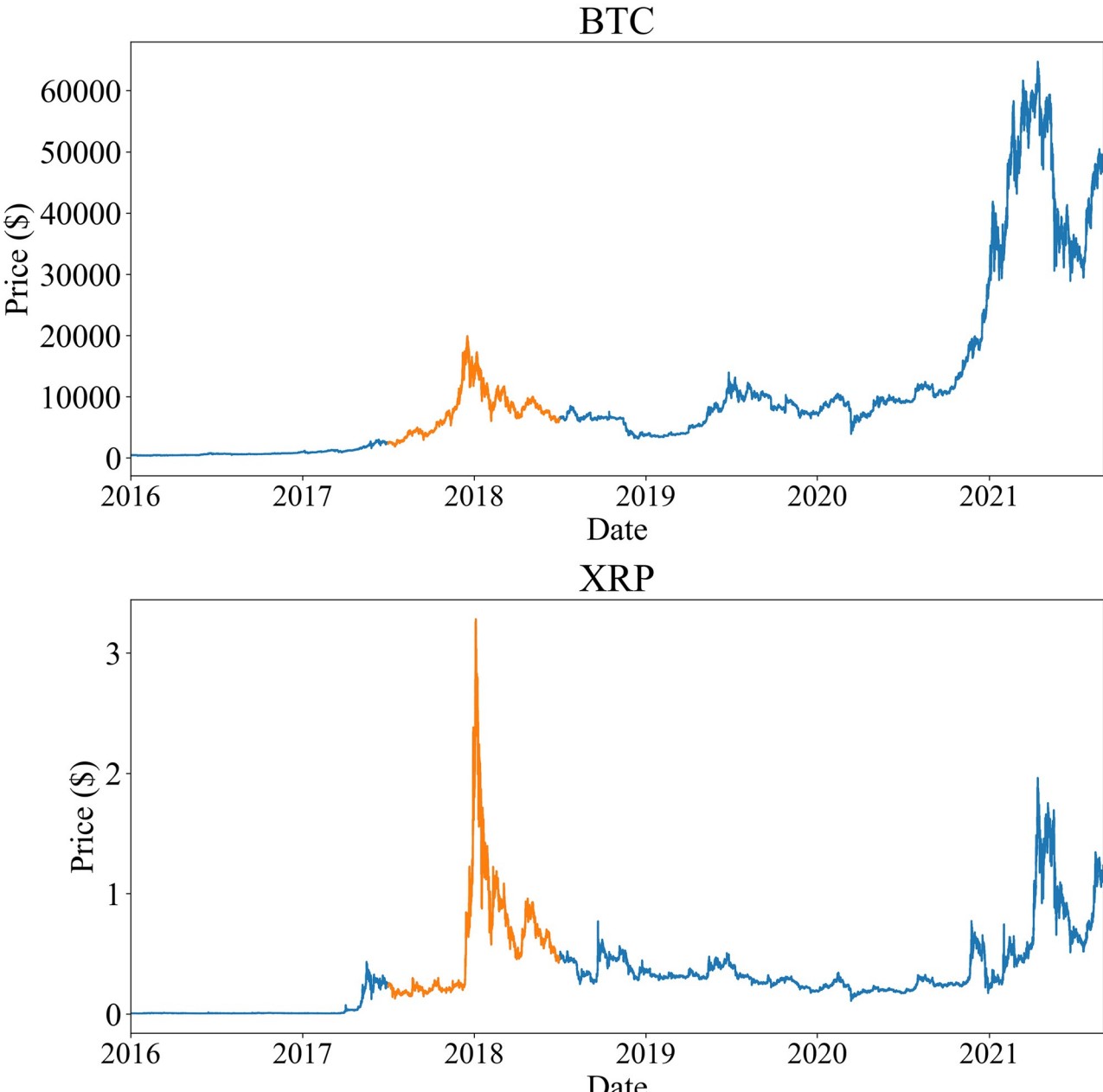

**Fig 1. Price chart for Bitcoin (top) and XRP (bottom).** Orange lines represent the price for the period from July 2 2017 to June 30 2018. The peaks are located at December 17 2017 for Bitcoin and January 4 2018 for XRP.

Bitcoin and January 4 2018 for XRP. We will focus on the period of one year around these peaks, which is made of the six months before and the six months after the peak of the bubble (July 2 2017 Sunday to June 30 2018 Saturday), shown by the orange line in Fig 1. We call the period **the target period**, which we study in this paper.

In either case of Bitcoin and XRP, each transaction is a transfer of crypto asset from one player to another with a certain amount. Our interest lies in the time-scale of days and weeks, for which the price of cryptoassets change significantly, it suffices to aggregate the transactions and to sum up the amounts of transfer over a relatively long time-scale. As we shall focus on "regular players" based on the frequency of transactions, most of them are doing business activities such as Exchanges and Services. In fact, there exists a daily periodicity, namely 24 hours depending on the geographical region where the player is located around the globe (see [23, Sec. 2] for example). Therefore, the most natural choice to aggregate the transactions would be daily aggregation, that is, to aggregate the transactions in each day based on the time-stamp (in the case of Bitcoin) and executed time (in the case of XRP) of each transaction.

It is straightforward to construct the daily transaction of crypto assets for each pair of players from our data sets. A daily network comprises of players as nodes and aggregated transfers as directed links. Each node has incoming and outgoing flow of crypto assets. We denote by $f_d^{(\text{in})}$ and $f_d^{(\text{out})}$ the total amount of **incoming** daily transfer and that of **outgoing** one, on day $d$, respectively. $d$ corresponds to one of the days during the target period.

One additional point to be considered is the fact that those business activities have a weekly periodic behavior for an obvious reason. That is to say, more transactions take place in week-days than in weekends. Let us denote by $t$ the index of week (we define a week to start on a Sunday). As we shall examine different short periods in the target period, we will consider a period from a week $t$ to $t + T$, denoted by $(t, T)$. For example, $(t = 1, T = 4)$ means the four weeks from the first week of the target period. The entire target period of one year is then $(t = 1, T = 52)$. When a particular day $d$ belongs to a period $(t, T)$, we denote this as $d \in (t, T)$. Thus if $d$ belongs to one week of $t$, we have $d \in (t, T = 1)$.

Now let us define a set of **active players** for a given period $(t, T)$ as the players who made transaction at least once during the period. We denote the set by $P_{(t, T)}$. In other words, a player belongs to $P_{(t, T)}$, if the player satisfies $f_d^{(\text{in})} > 0$ or $f_d^{(\text{out})} > 0$ for one or more $d \in (t, T)$. For the target period, the set of active players $P_{(t = 1, T = 52)}$, or $P_{(1,52)}$ for short, is denoted by $P_{\text{active}}$. In order to focus on "regular players" among $P_{\text{active}}$, we then define **regular players** precisely by the following

$$P_{\text{reg}} = \bigcap_{t=1}^{52} P_{(t,1)}, \tag{1}$$

that is, those players who made transaction *every week* at least once all over the target period. Obviously, $P_{\text{reg}} \subseteq P_{\text{active}}$. In the following, we use the notation $|P|$ to represent the number of elements of the set of $P$.

The numbers of the active players and the regular players for Bitcoin are $|P_{\text{active}}^{\text{BTC}}| = 65,823,109$ and $|P_{\text{reg}}^{\text{BTC}}| = 1,097$ respectively. In the case of XRP, they are $|P_{\text{active}}^{\text{XRP}}| = 950,749$ and $|P_{\text{reg}}^{\text{BTC}}| = 32$, respectively, both of which are in the 2–3% range of the corresponding values of Bitcoin. Table 1 summarizes these numbers as well as the numbers for $P_{(1,4)}$ for the first 4-week period to be used later. These regular players are presumably "big players" who play dominant roles in the transaction network, which we shall study in what follows.

**Table 1. Number of participating players.**

| The periods | Bitcoin | XRP |
|---|---|---|
| Active players, $P_{\text{active}}$, | 65, 823, 109 | 950, 749 |
| Players in the first 4-week period, $P_{(1,4)}$ | 5, 224, 054 | 39, 868 |
| Regular players, $P_{\text{reg}}$ | 1, 097 | 32 |

## Method: Flow-weighted frequency

In the preceding section, we defined the players $P_{(t, T)}$ for a given period $(t, T)$ and the regular players $P_{\text{reg}}$ for the entire target period. While those players can have important roles with respect to the transaction network, we need to capture different features in the roles. First, one can easily expect that outgoing and incoming flows of crypto assets vary among the players depending on what kind of activity such a regular player is doing. For example, a player doing the business activity of Exchanges may not want to take a position of surplus or deficit, because the usual behavior of highly volatile crypto price (recall Fig 1) can quite easily affect the net value of the player's total asset. It is likely that such a player would take a "balanced" position as much as possible. Another example is players doing the activity of mining in addition to Exchanges or Services. They can have the excess flow of outgoing cryptoassets because of the very nature of miners. These obvious examples suggest us to distinguish the outgoing crypto asset and the incoming one of each player.

More important is to take into account the amount of flow when we define how frequent a player appears in the transaction network. Suppose we have a daily sequence of outgoing and incoming flow for a player during a given period $(t, T)$:

$$\boldsymbol{f}^{(\text{out})} \quad = \{f_d^{(\text{out})}\} \quad \text{where } d \in (t, T), \tag{2}$$

$$\boldsymbol{f}^{(\text{in})} \quad = \{f_d^{(\text{in})}\} \quad \text{where } d \in (t, T), \tag{3}$$

where $f_d^{(\text{out})} > 0$ and $f_d^{(\text{in})} > 0$ represent the amount of outgoing flow and that of incoming flow for day $d$ during the period respectively.

To define an "effective" frequency with which a player is doing daily transaction, with respect to outgoing and incoming flow, we propose the following formula:

$$(A^{(\text{out})}, A^{(\text{in})}) = \frac{1}{\text{Max}(\boldsymbol{f}^{(\text{out})}, \boldsymbol{f}^{(\text{in})})} \left( \text{Total}(\boldsymbol{f}^{(\text{out})}), \text{Total}(\boldsymbol{f}^{(\text{in})}) \right), \tag{4}$$

where "Total" is the total amount of flow during the period, and "Max" is the maximum of daily amount. (One of the authors (H.A.) defined a similar but different index in the paper [25], called "Flow index". The index was motivated by the inverse of Herfindahl-Hirschman index. The present index is simpler than the one in [25], but can quantify an effective frequency properly). We call the indices, $A^{(\text{out})}$ and $A^{(\text{in})}$, **flow-weighted frequencies** of the player for outgoing and incoming flows respectively, and shall use the abbreviation of **F-frequency**. The indices obviously depend on the period $(t, T)$. We will explicitly display the dependence by using the notation of $A^{(\text{out})}(t, T)$ and $A^{(\text{in})}(t, T)$. Note that the F-frequency is invariant under the change of temporal order of flows, and also under the change of the scale in the amount.

To understand why this index captures the frequency by taking into account of the amount of flow, let us consider an illustrative example:

$$\boldsymbol{f}^{(\text{out})} = \{10^4, 0, 1, 50, 0, 0, 0\}, \tag{5}$$

$$\boldsymbol{f}^{(\text{in})} = \{0, 1, 200, 0, 0, 0, 0\}, \tag{6}$$

each with 7 daily amounts of flows corresponding to 7 days in a week $T = 1$. For this example, we have

$$(A^{(\text{out})}, A^{(\text{in})}) = (1.0051, 0.0201). \tag{7}$$

One can see that these values correspond to our intuitive quantification of frequency because the dominant flow takes place mostly on the first day in the incoming flow, while the outgoing flow is quite small compared with the incoming one.

For another illustration, consider the case

$$\boldsymbol{f}^{(\text{out})} = \{\underbrace{a, a, \cdots, a}_{m \text{ times}}, 0, \cdots, 0\}, \tag{8}$$

$$\boldsymbol{f}^{(\text{in})} = \{\underbrace{b, b, \cdots, b}_{n \text{ times}}, 0, \cdots, 0\}, \tag{9}$$

for $a, b > 0$ and the condition that $a \gg b$. Then we have

$$(A^{(\text{out})}, A^{(\text{in})}) = \left(m, \frac{b}{a} n\right) \simeq (m, 0) \tag{10}$$

One can see that the F-frequency indices give us a reasonable quantification for the frequency, taking into account how a player has a role in both of outgoing and incoming flows.

We note that from the definition (4) it immediately follows that the maximum possible value that each of $A^{(\text{out})}$ and $A^{(\text{in})}$ for a given period $(t, T)$ can take is given by the number of days contained in the period, that is, $7 \times T$.

## Results

### The three-branch structure

To see a typical structure of F-frequency before the sharp price increases, we show the scatter plots of the F-frequency for the first four-week period from Sunday, July 2 to Saturday, July 29, 2017 in Fig 2. Red points represent the F-frequency of the regular players. The number of Bitcoin and XRP active players for the four-week period are $|P_{(1,4)}^{\text{BTC}}| = 5,224,054$, and $|P_{(1,4)}^{\text{XRP}}| = 39,868$ players, respectively. They are much larger than the number of regular players:$|P_{\text{reg}}^{\text{BTC}}| = 1,097$ and $|P_{\text{reg}}^{\text{BTC}}| = 32$. By definition of the F-frequency (4), the maximal value is $\mathbf{A} = (28, 28)$, and there are no points in $A^{(\text{out})} < 1$ and $A^{(\text{in})} < 1$. Both plots in Fig 2 show characteristic behavior of F-frequency, whose points are concentrated on three regions $A^{(\text{out})} \simeq A^{(\text{in})}$, $A^{(\text{out})} \sim 0$, and $A^{(\text{out})} \sim 0$ in addition to the region around the origin, $A^{(\text{out})} < 5$ and $A^{(\text{in})} < 5$ for Bitcoin, and $A^{(\text{out})} < 3$ and $A^{(\text{in})} < 3$ for XRP. We call this behavior the **three brunch structure (TBS)**.

To classify players to reflect the TBS for a given period $(t, t + T)$, we define the set of "In-players" $P_{(t, T)\text{In}}$, "Balanced-players" $P_{(t, T)\text{Bal}}$, and "Out-players" $P_{(t, T)\text{Out}}$ by dividing the set of

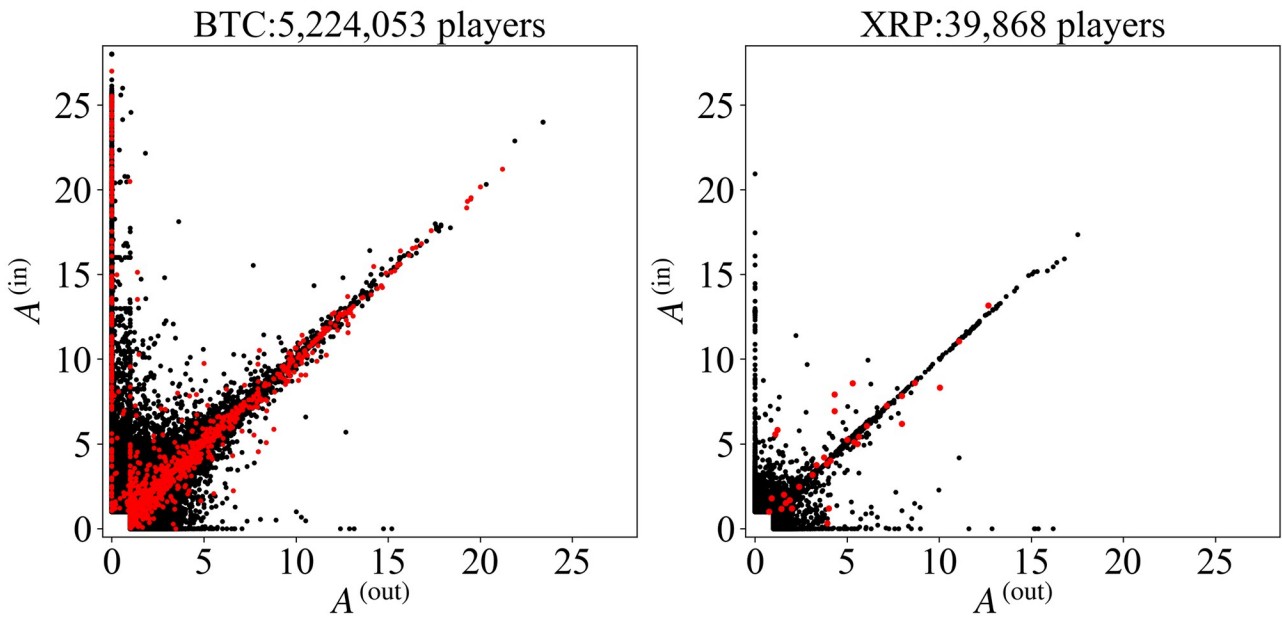

**Fig 2. Bitcoin (left) and XRP (right) players in the two-dimensional space of F-frequency for the first four-week period, $(t, T) = (1, 4)$, from Sunday, July 2 to Saturday July 29, 2017.** Red points represent the F-frequency of the regular players. The points of the regular players of XRP are plotted twice as large as the other points to increase visibility.

points of F-frequency into three regions of 30 degrees each as

$$P_{(t,T)\text{In}} := \left\{ i \in P_{(t,T)} \middle| A_i^{(\text{in})} \geq A_i^{(\text{out})} \tan \frac{\pi}{3} \right\}, \tag{11}$$

$$P_{(t,T)\text{Bal}} := \left\{ i \in P_{(t,T)} \middle| A_i^{(\text{in})} \geq A_i^{(\text{out})} \tan \frac{\pi}{6}, A_i^{(\text{in})} < A_i^{(\text{out})} \tan \frac{\pi}{3} \right\}, \tag{12}$$

$$P_{(t,T)\text{Out}} := \left\{ i \in P_{(t,T)} \middle| A_i^{(\text{in})} < A_i^{(\text{out})} \tan \frac{\pi}{6} \right\}. \tag{13}$$

For Bitcoin, the number of In-, Bal-, and Out-players for the first four-week period are $|P_{(1,4)\text{In}}^{\text{BTC}}| = 1,030,547$, $|P_{(1,4)\text{Bal}}^{\text{BTC}}| = 3,780,847$, and $|P_{(1,4)\text{Out}}^{\text{BTC}}| = 412,659$, respectively. In contrast, their regular players are $|P_{(1,4)\text{In}}^{\text{BTC}} \cap P_{\text{reg}}^{\text{BTC}}| = 263$, $|P_{(1,4)\text{Bal}}^{\text{BTC}} \cap P_{\text{reg}}^{\text{BTC}}| = 759$, and $|P_{(1,4)\text{Out}}^{\text{BTC}} \cap P_{\text{reg}}^{\text{BTC}}| = 75$. On the other hand, for XRP, the number of In-, Bal-, and Out-players for the first four-week period are $|P_{(1,4)\text{In}}^{\text{XRP}}| = 22,847$, $|P_{(1,4)\text{Bal}}^{\text{XRP}}| = 8,827$, and $|P_{(1,4)\text{Out}}^{\text{XRP}}| = 8,194$; and their regular players are $|P_{(1,4)\text{In}}^{\text{XRP}} \cap P_{\text{reg}}^{\text{XRP}}| = 4$, $|P_{(1,4)\text{Bal}}^{\text{XRP}} \cap P_{\text{reg}}^{\text{XRP}}| = 26$, and $|P_{(1,4)\text{Out}}^{\text{XRP}} \cap P_{\text{reg}}^{\text{XRP}}| = 2$, respectively. These are summarized in Table 2

We will discuss how these players change their positions during the target period, in particular around the peak of the price.

**Table 2. The number of In-, Bal-, and Out-active and regular players for the first four-week period.**

|  | Active players in the first 4-week period | | | Regular players | | |
|---|---|---|---|---|---|---|
|  | **In** | **Bal** | **Out** | **In** | **Bal** | **Out** |
| Bitcoin | 1, 030, 547 | 3, 780, 847 | 412, 659 | 263 | 759 | 75 |
| XRP | 22, 847 | 8, 827 | 8, 194 | 4 | 26 | 2 |

## Identified players in the three branches

In the case of Bitcoin, for the players who are doing business activities such as Exchanges, Services, Gambling, and mining, it is known that one can obtain the identity of users. Such information could be useful for our study, even if not exhaustive. In fact, one can register to service, make transactions, and watch which wallet Bitcoins were merged with, or which wallet it was withdrawn from. This straightforward but laborious method of identification has been done by curious individuals and by investigating agencies. We employ one of the well-known web site, `WalletExplorer.com` [27]. Which provides a comprehensive list of such identification with the classification of business activities. See Supporting Information S1 File. for details. Classification of business activities is given by five categories of Exchanges, Services, Gambling, Mining Pools, and Old/Historic. As a result, we have a list of 366 identified players. We denote the set of identified players as $P_{\text{ID}}$.

We define the set of identified active and regular players by the intersections:

$$P_{\text{active}}^{\text{ID}} = P_{\text{active}}^{\text{BTC}} \cap P_{\text{ID}}, \tag{14}$$

$$P_{\text{reg}}^{\text{ID}} = P_{\text{reg}}^{\text{BTC}} \cap P_{\text{ID}}. \tag{15}$$

There are $|P_{\text{active}}^{\text{ID}}| = 366$ (of $65, 823, 109$ active players $= |P_{\text{active}}^{\text{BTC}}|$) active players and $|P_{\text{reg}}^{\text{ID}}| = 63$ (of $1, 097$ regular players $= |P_{\text{reg}}^{\text{BTC}}|$) regular players. (Multiple wallets may be assigned to the same player. There are two such players in our analysis: The categories of wallets of one player are Old/Historic and Exchanges categories. We assigned the player to the Exchanges category because the wallet of Old/Historic is presumed to be inactive. The categories of wallets of the other player are all Exchanges so that we assigned him the Exchange category). Similarly, we define the identified players for a period $(t, t + T)$ by

$$P_{(t,T)}^{\text{ID}} = P_{(t,T)}^{\text{BTC}} \cap P_{\text{ID}}. \tag{16}$$

In addition to the above mentioned classification of business activities, we can classify players according to whether a player made mining during the target period. As a result, the identified players can be classified into ten categories. Table 3 summarizes the results of these classifications. There are no mining players in the Gambling category. This behavior is natural since the purpose of business activity of Gambling is not mining. The players in the Old/Historic category do not seem to be mining either, which can be understood as they are not so active. In fact, there is no regular player in the Old/Historic category. Players in Exchange, Pools, and Services/Others do mining. The pools have the highest ratio of mining, as expected from the name of the category. Some players in Exchange and Services/Others might be

Table 3. Classification of identified active and regular players. "Mining" is "Yes" if a player mined Bitcoins during the target period, and "No" otherwise.

| | Active players, $P_{\text{active}}^{\text{ID}}$ | | Regular players, $P_{\text{reg}}^{\text{ID}}$ | |
|---|---|---|---|---|
| Mining | Yes | No | Yes | No |
| Exchanges | 14 | 78 | 12 | 27 |
| Gambling | 0 | 32 | 0 | 8 |
| Old/Historic | 0 | 46 | 0 | 0 |
| Pools | 3 | 5 | 1 | 0 |
| Services/Others | 7 | 35 | 7 | 8 |
| Total | 24 | 342 | 20 | 43 |

**Table 4. Category of identified active and regular players appeared in the first four-week period from July 2 to July 29, 2017.** "Mining" is "Yes" if a player mined Bitcoins during the target period, and "No" otherwise.

| | Active players, $P^{\text{ID}}_{(1,4)}$ | | | | | | Regular players, $P^{\text{ID}}_{\text{reg}}$ | | | | | |
| | In | | Bal | | Out | | In | | Bal | | Out | |
| Mining | Yes | No | Yes | No | Yes | No | Yes | No | Yes | No | Yes | No |
|---|---|---|---|---|---|---|---|---|---|---|---|---|
| Exchanges | 0 | 17 | 12 | 38 | 1 | 5 | 0 | 1 | 11 | 25 | 1 | 1 |
| Gambling | 0 | 11 | 0 | 18 | 0 | 1 | 0 | 0 | 0 | 8 | 0 | 0 |
| Old/Historic | 0 | 29 | 0 | 4 | 0 | 4 | 0 | 0 | 0 | 0 | 0 | 0 |
| Pools | 0 | 2 | 2 | 0 | 0 | 0 | 0 | 0 | 1 | 0 | 0 | 0 |
| Services/Others | 0 | 12 | 7 | 9 | 0 | 2 | 0 | 1 | 7 | 7 | 0 | 0 |
| Total | 0 | 71 | 21 | 69 | 1 | 12 | 0 | 2 | 19 | 40 | 1 | 1 |

providing mining services or do mining by themselves. The ratio of regular players who do mining is higher than that of active players.

To see where these players belong in the three-branch structure, we further classify them using the data of the F-frequency for the first four-week period from July 2 to July 29, 2017. We find the number of identified active player is $|P^{\text{ID}}_{(1,4)}| = 174$, which is smaller than $|P^{\text{ID}}_{\text{active}}| = 366$ because not all identified active players made transactions during the first four-week period. Since the all regular players are active during the target period, the number of identified regular players is the same as the total identified regular players, $|P^{\text{ID}}_{\text{reg}}| = 63$. The result is shown in Table 4. Mining players are more likely to send their mined bitcoins elsewhere, which increases their $A^{(\text{out})}$. As a result, the probability of staying in the Bal- or Out-regions will increase. All players belonging to the pool category do mining, and are Bal-players. We can see that most of the identified players belong to Bal-branch for regular players.

## Time-series of thee-brunch structure

First, let us show the time series of the number of Bitcoin players on a weekly basis in the left panel of Fig 3, which are given by $|P^{\text{BTC}}_{(t,1)i}|$, where $i \in \{\text{In, Bal, Out}\}$. As price data, we employ

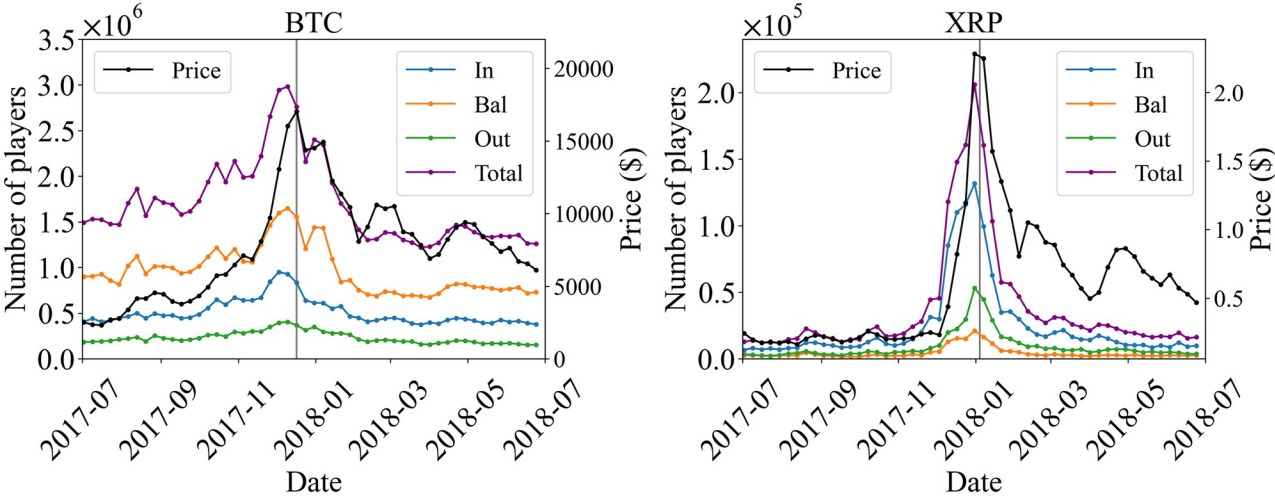

**Fig 3. Time series of the number of In-, Bal-, and Out-players (weekly total), $|P_{(t,1)\text{In}}|$, $|P_{(t,1)\text{Bal}}|$, and $|P_{(t,1)\text{Out}}|$, for Bitcoin (left) and XRP (right).** The black line represents the one-week-average price. The week that contains the day of the peak, $t = 25$ is represented by the gray vertical line.

**Table 5. Correlation between the time difference of price, $\Delta p(t)$, and the time difference of the number of Total, In, Bal, and Out active players, $\Delta N_i(t)$.** $k$ represents the lag time. For example, $k = 1$ means the correlation between $\Delta p(t - 1)$ and $\Delta N_i(t)$.

| | Active players (BTC) | | | | Active Players (XRP) | | | |
|---|---|---|---|---|---|---|---|---|
| | **Total** | **In** | **Bal** | **Out** | **Total** | **In** | **Bal** | **Out** |
| $k = 0$ | 0.63 | 0.52 | 0.60 | 0.51 | 0.67 | 0.54 | 0.64 | 0.86 |
| $k = -1$ | 0.49 | 0.51 | 0.39 | 0.50 | 0.52 | 0.53 | 0.40 | 0.44 |
| $k = 1$ | 0.03 | 0.07 | 0.01 | 0.03 | 0.08 | 0.00 | 0.08 | 0.26 |

the price at 00:00:00 UTC as the price of the day $p_d$, and plot one-week-average price $p(t) := \sum_{d=0}^{6} p_{t+d}/7$. As the price (black line) approach the peak, the number of players increases, reaching a maximum just before the price peak. All In-, Bal- and Out-players behave similarly. The behavior of the total number is dominated by the Bal-players because the number of Bal-players is larger than others. In addition to the maximum peak, several bumps can be seen. This is thought to be correlated to the price. Similarly, for XRP in Fig 3, a sharp peak is observed just a little before the price peak, which is much sharper than that of Bitcoin. This is consistent with the fact that the price peak of XRP in Fig 1 is sharper than that of BTC. One characteristic difference is that Bitcoin has Bal-, In-, and Out-players in descending order of number, while XRP has In-, Out-, and Bal-players in descending order. Table 5 shows the correlation between the time difference of price, $\Delta p(t) := p(t + 1) - p(t)$, and the time difference of the number of Total, In, Bal, and Out players, $\Delta N_i := (t)N_i(t + 1) - N_i(t)$ ($i = $ In, Bal, Out), which are defined as

$$\rho_i(k) := \frac{\sum_{t=1}^{51}(\Delta p(t - k) - \overline{\Delta p})(\Delta N_i(t) - \overline{\Delta N_i})}{\sqrt{\left(\sum_{t'=1}^{51}\left(\Delta p(t' - k) - \overline{\Delta p}\right)^2\right)\left(\sum_{t''=1}^{51}\left(\Delta N_i(t'') - \overline{\Delta N_i}\right)^2\right)}}, \tag{17}$$

where $\overline{\Delta p} := \sum_{t=1}^{51} \Delta p(t)/51$, and $\overline{\Delta N_i} := \sum_{t=1}^{51} \Delta N_i(t)/51$. Here, $k$ represents the lag time.

There are correlations for $k = 0$ and $k = -1$ in both BTC and XRP, while no correlation is seen for $k = 1$. These correlations suggest that the price changes follow the changes in the number of users in this target period. The comparison with other periods will be discussed in the discussion section.

Second, we show the time series of the number of In-, Bal-, and, Out-regular players in Fig 4. These are given by $|P_{(t,1)i} \cap P_{reg}|$ ($i \in$ {In, Bal, Out}). Unlike in the previous case, the total number of players does not change. The only ratio changes. For Bitcoin, the numbers of In- and Bal-players are in the same order, while the number of Out-players is smaller compared with others. We can see the clear structure at the price peak. From the week before the peak to the week of the peak, the number of In-players increases significantly while the number of Bal-players decreases. The ratio of Bal-players is larger before the peak, but after the peak, the ratio of In-players is larger.

The situation is different for XRP. The number of Bal-players is typically larger than those of In- and Out-players, except at the beginning of the target period. Unlike the Bitcoin case, it seems to be difficult to observe clear behavior due to the small number of regular players ($|P_{reg}^{XRP}| = 32$). The correlation for regular players is shown in Table 6.

Third, we show the scatter plots of F-frequency for three weeks around the peak in Fig 5. Unfortunately, in both Bitcoin and XRP cases, it doesn't seem easy to read quantitative behavior from this data. To see the detailed behavior around the price peak, let us define the

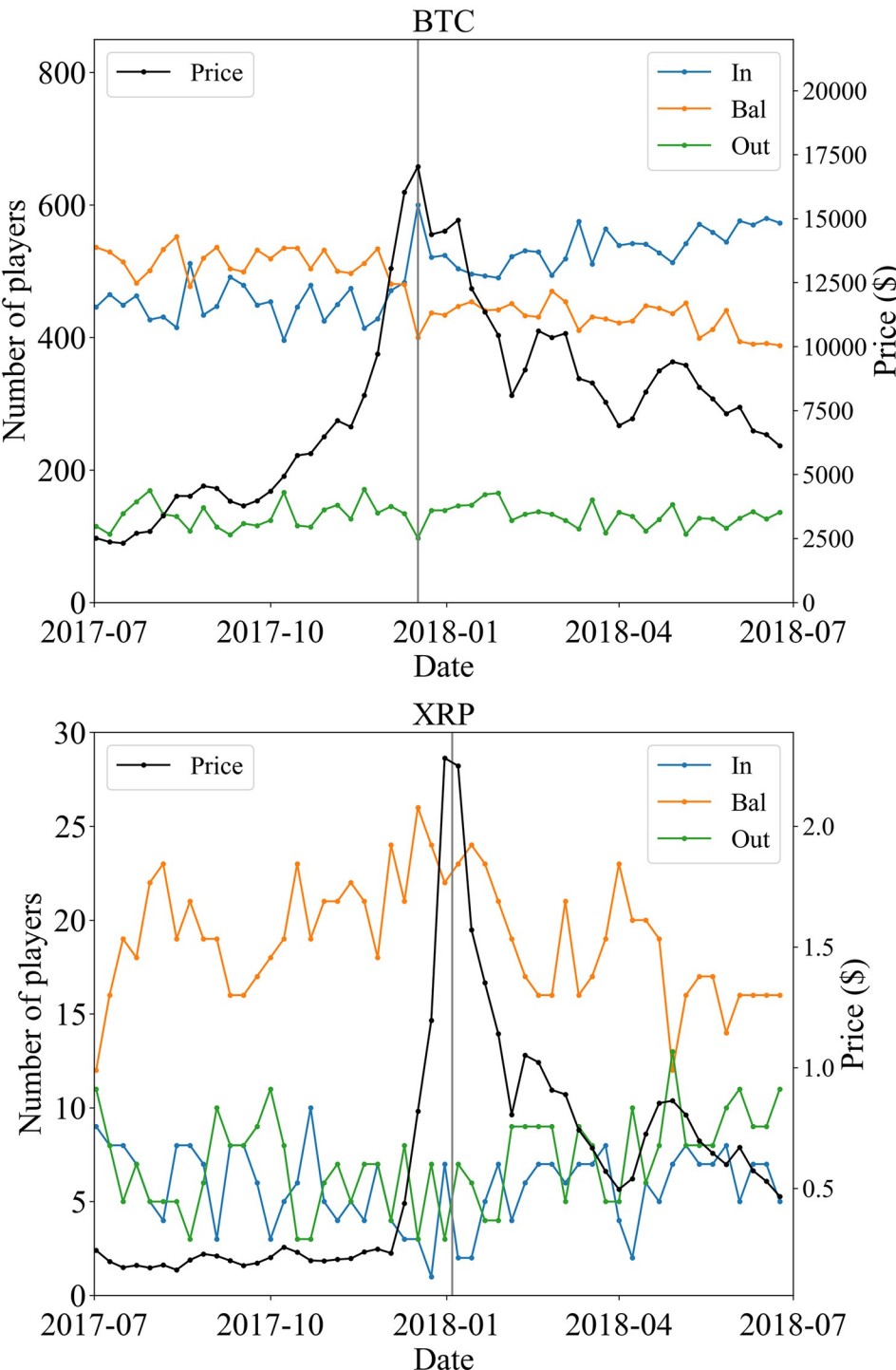

**Fig 4. Number of In- (blue), Bal- (orange), and Out- (green) regular players, $|P_{(t,1)i} \cap P_{reg}|$ ($i \in$ {In, Bal, Out}), for Bitcoin (top) and XRP (bottom).** The black line represents the one-week-average price. The week that contains the day of the peak, $t = 25$ is represented by the gray vertical line.

**Table 6. Correlation between the time difference of price, $\Delta p(t)$, and the time difference of the number of Total, In, Bal, and Out regular players, $\Delta N_i(t)$.** $k$ represents the lag time. For example, $k = 1$ means the correlation between $\Delta p(t-1)$ and $\Delta N_i(t)$.

| | Regular players (BTC) | | | Regular players (XRP) | | |
|---|---|---|---|---|---|---|
| | **In** | **Bal** | **Out** | **In** | **Bal** | **Out** |
| $k = 0$ | 0.09 | −0.11 | −0.02 | 0.23 | −0.07 | −0.14 |
| $k = -1$ | −0.14 | 0.15 | 0.05 | −0.09 | −0.07 | 0.14 |
| $k = 1$ | 0.24 | −0.14 | −0.22 | −0.18 | 0.02 | 0.14 |

transition rate from region $j$ at the time $t$ to $i$ at $t + \Delta t$ as

$$W_{ji}(t) := \frac{1}{N_{\text{tot}}}\left(N_j(t + \Delta t) - N_i(t)\right). \tag{18}$$

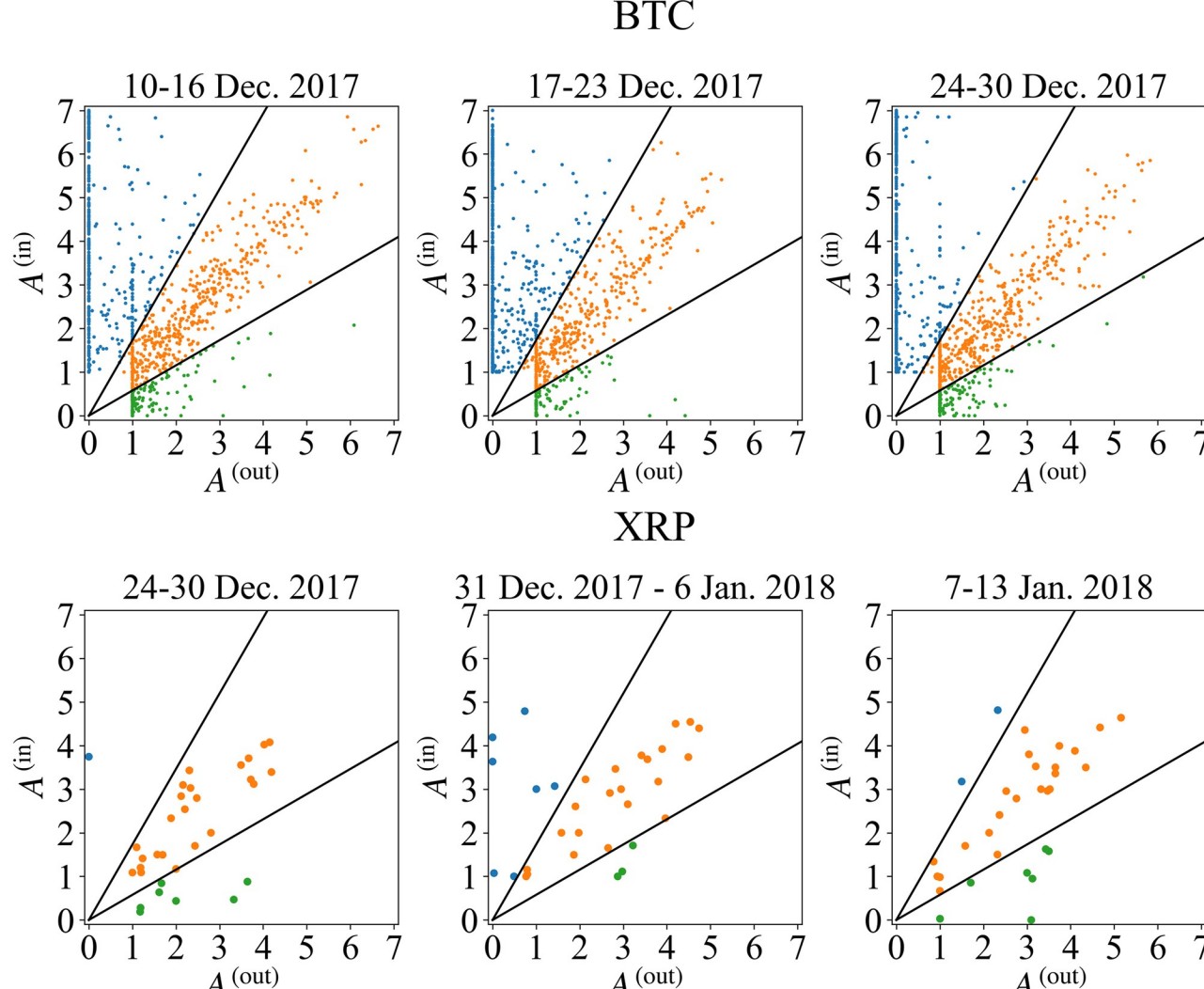

**Fig 5. F-frequency for Bitcoin (top) and XRP (bottom) regular players around the peak price day.** Blue, orange, and green points represent In-, Bal-, and Out-players, respectively. Black lines divide these regions.

Here $N_i(t) = |P_{(t,1)i} \cap P_{\mathrm{reg}}|$ ($i \in$ {In, Bal, Out}), and $N_{\mathrm{tot}} = |P_{\mathrm{reg}}|$, are the number of $i$ and total regular players. Fig 6 shows the heatmaps of the transition rate around the price peak. Each component represents a transition from a column element to a row element. The heatmap of $10 \rightarrow 17$ Dec. 2017 for Bitcoin (right top of BTC in Fig 6) shows a large net inflow, $W_{\mathrm{In,Bal}} - W_{\mathrm{Bal,In}} = 0.084$, from the Bal-branch to the In-branch, which is relatively larger than other net transitions, and it causes the characteristic structure in Fig 4. For XRP, we can see a relatively large net inflow from the Bal-branch to the Out-branch, $W_{\mathrm{Out,Bal}} - W_{\mathrm{Out,Bal}} = 0.126$ in the transition $17 \rightarrow 24$ Dec. 2017 (left top of XRP in Fig 6). However, it does not seem to make a significant characteristic because the other transitions are not so small in comparison.

Finally, we show the plots of the number of In-, Bal-, and Out-identified regular players categorized in Exchange, Gambling, and Service/Others, in Fig 7. We have not plotted the time series of the number of identified regular players categorized as Pool. This is because the only one identified regular player is categorized as the Pool, and it is located in the Bal-branch during the target period. In all categories, the Bal-players dominate the other players. For identified regular players categorized as Gambling and Services/Others, there is no behavior that should be pointed out. On the other hand, for players categorized in Exchanges, the number of In- (Bal-) and players decreases (increases) in time. To see the role of the identified regular users in the characteristic behavior in Fig 4, we plot the time series of the number of regular non-identified and identified In-, Bal-, and Out-players in Fig 8. The figure shows the characteristic behavior caused by non-identified regular users.

## Discussion

### Interpretation of three-branch structure

In Fig 2, we found that there exist three groups of active and regular players in both of the cases for Bitcoin and XRP, as is evident in a three-branch structure.

The branch along the diagonal line, for which the F-frequencies $A^{(\mathrm{out})}$ and $A^{(\mathrm{in})}$ are equal, corresponds to the players who have a balance between outgoing flow and incoming flow. This branch (Bal) comprises of those players who regularly balance between surplus and deficit of cryptoassets on the daily basis with respect to flow-weighted frequencies. For example, regular players doing the business activities of Exchanges would not dare to take an unbalanced position, either of surplus and deficit of cryptoassets, simply because such a position can be extremely risky under the volatile asset price of the crypto; the player may lose the chance of rising asset price or may experience the risk of falling asset price. Actually, in the case of Bitcoin, for which we have partial information on the identity of players, one can see from Table 4 that most of the active and regular players in the Bal-branch are Exchanges.

Another branch along the vertical axis (In), for which $A^{(\mathrm{out})} \simeq 0$, corresponds to the players who are accumulating the cryptoasset regularly on the daily basis. Presumably these players have a policy favorable for taking a position of surplus cryptoasset in the anticipation that the asset price rises in the future, or that even after the crash the price might possibly revert to its previous level. Such players exist among Exchanges, and existed in the past among Old/Historic players, as one can see from Table 4. It is interesting to observe that the values of $A^{(\mathrm{in})}$ can be relatively large, compared with the maximally possible value, indicating that the activity is quite strong in terms of flow-weighted frequency.

The other branch along the horizontal axis (Out), for which $A^{(\mathrm{in})} \simeq 0$, corresponds to the players who are providing outgoing flows. Note that the values of $A^{(\mathrm{out})}$ are much smaller than the maximally possible value. In the case of Bitcoin, it would be reasonable that this branch includes miners, or such players who do the activity of mining in addition to the business of Exchanges. Actually, according to Table 3 there exists a certain number of Exchanges who do

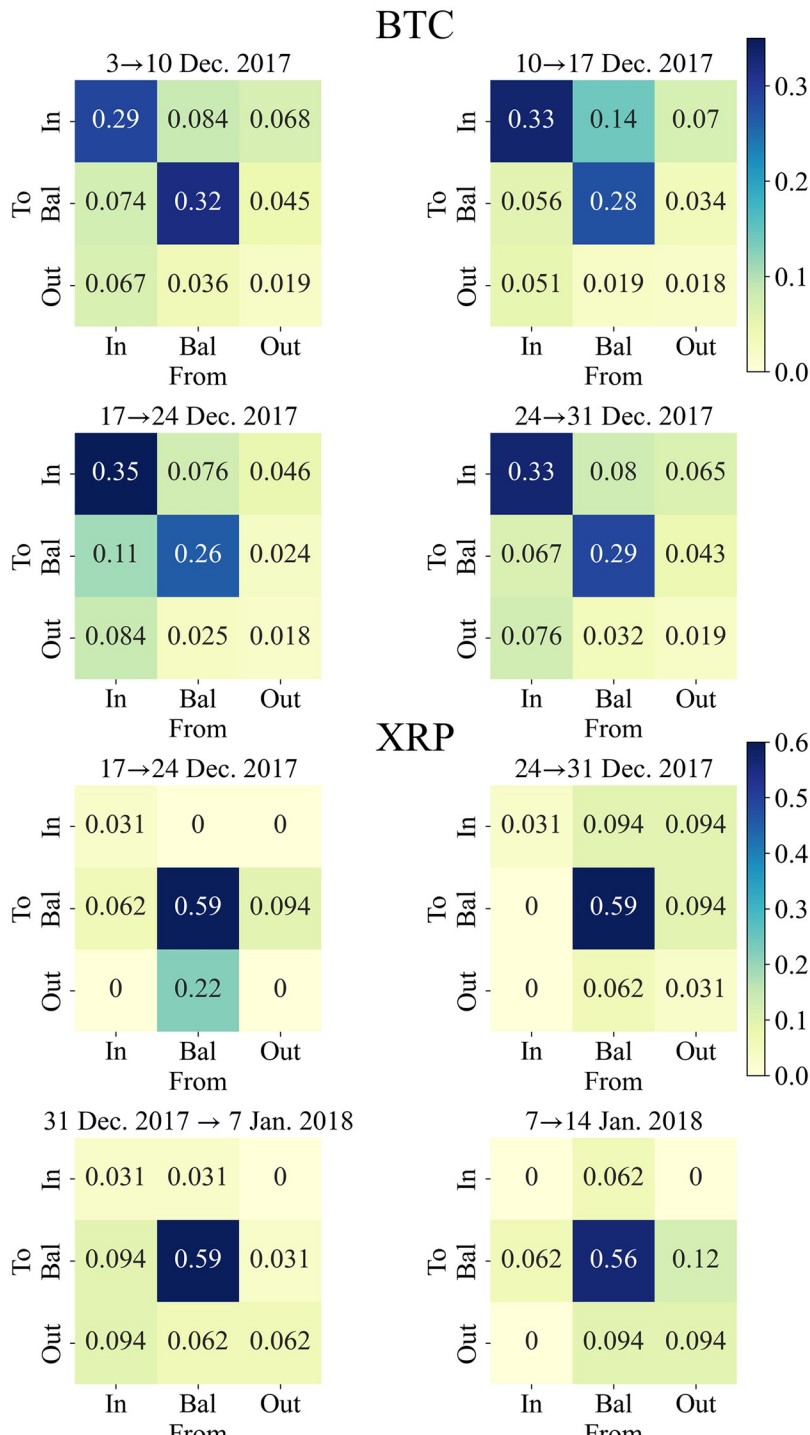

**Fig 6. Heatmaps of transition rates around the price peak for Bitcoin (top) and XRP (bottom).** Each component represents a transition from a column (From) element to a row (To) element. $n \rightarrow m$ Dec. 2017 means the transition rate from the week starting at $n$ to at $m$ Dec. 2017.

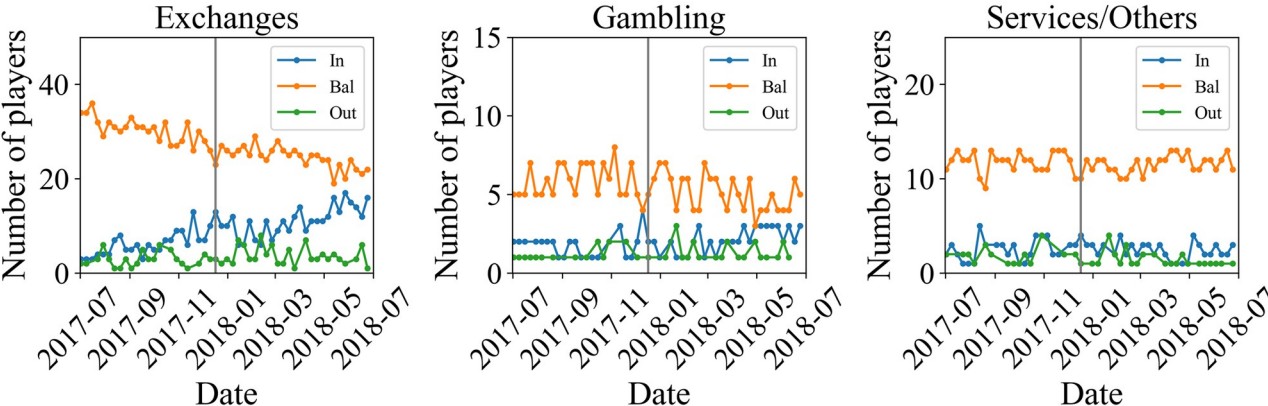

**Fig 7. Number of identified regular In-, Bal-, and Out-players for Bitcoin categorized as exchange (left), gambling (middle), and service/others (right).** In-, Bal-, and Out-players are plotted in blue, orange, and green, respectively. The gray vertical line represents the week that contains the day of the peak.

the mining in their business activity. An obvious incentive to do the mining is a reward of crypto that yields an increasing amount of cryptoasset, especially in the bubble phase of asset price, even if the miner pays a lot for winning among the competing peers.

## Temporal change of the three-branch structure and price

Fig 3 obviously tells us that the numbers of players in the three branch of Bal, In, and Out and also in the total number is highly correlated with the bubble and crash of the price, both for Bitcoin and XRP. In the case of Bitcoin, comparing the two phases, one before the crash and the other after it, one can see that each number of players did not revert to its level before the bubble. On the other hand, in the case of XRP, the number of players did not change much, or even increased slightly. The surge of the number of participating players is reasonable in the

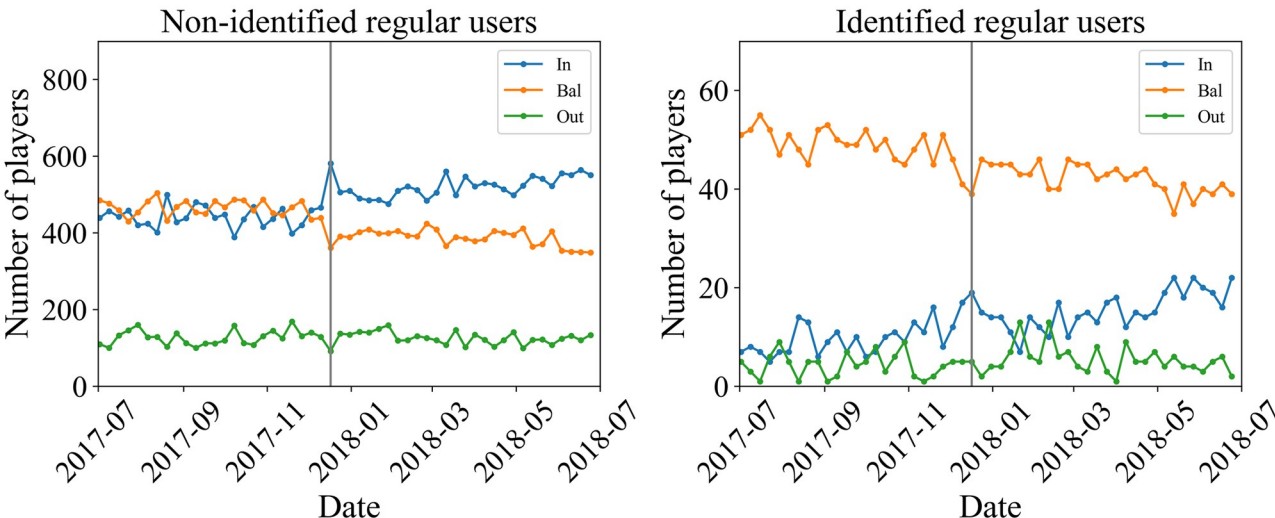

**Fig 8. Number of non-identified (left) and identified (right) regular In-, Bal-, and Out-players for Bitcoin.** In-, Bal-, and Out-players are plotted in blue, orange, and green, respectively. The gray vertical line represents the week that contains the day of the peak.

sense that more and more players participated during the bubble and abruptly exited after the crash.

It is also interesting, as we explained in the Results regarding Fig 3, the surge of the numbers of players in all the branches precedes the peak of the price for both of Bitcoin and XRP. It is tempting to consider the preceding surge as a precursor that can be potentially detected by the network of transaction.

Regarding the regular players, Fig 4 leads us to an interesting observation. In the case of Bitcoin, the number of regular players in the Bal-branch was larger than that of regular players in the branch of In before the peak of price, but right after the peak the former became smaller than the latter. This regime switching, so to speak, is quite evident in the top panel of Fig 4. On the other hand, in the case of XRP, there was no such regime switching, even if the time-series for the regular players for each of the three branches were highly volatile as shown in the bottom panel of Fig 4.

We do not have a definite answer to the question why the two cases are dramatically different from each other, but can argue that the identity of regular players in the case of Bitcoin may give us a hint to the answer as follows.

From Fig 7, we can see that for regular players classified as Exchanges, there is a decreasing trend for the Bal-branch and an increasing trend for the In-branch overtime. We do not observe a clear trend for regular players classified as Gambling or Service/Other. Also, for any of the identified regulars in Bitcoin shown in Fig 7, neither the number of In-players nor the number of Bal-players shows any significant change at the price peak. This fact suggests that the abrupt change in the number of players at the price peak seen in the upper panel of Fig 4 for BTC regular players is due to a change in the number of non-identified regular players. In other words, this may be the effect of the influx of new players encouraged by the bull market. In fact, in Fig 8, there is a sharp change in the In- and the Bal-branch for the non-identified regular players at the price peak. In contrast, we do not observe a significant change for the identified regular players. For the XRP regular player shown in the lower panel of Fig 4, there is no significant change at the price peak. This behavior is similar to that of the identified regular player in Bitcoin.

Therefore, if we focus on those regular players who are known to the world to an extent such that they can be identified, the above mentioned regime switching is not observed; rather, the behavior of the temporal change of three-branch structure is quite similar to that for the case of XRP. In other words, the regime switching was presumably brought about by the regular players who are not necessarily dominant and stable in the case of Bitcoin, while such players are simply absent in the case of XRP. Unfortunately, we do not have even a partial information of identity of regular players for XRP, but can infer about this similarity between Bitcoin and XRP as far as the network of transaction and the bubble/crash price dynamics are concerned.

## Transitions among the three branches

In the case of Bitcoin, In-players increased significantly while the number of Bal-players decreased at the price peak. Fig 4 shows that the number of In- and Bal-players changed like a step function from December 2017 to January 2018. In other words, there is a clear correlation between the price change of crypto assets and the number of In- and Bal-players. Did the increase in In-players cause the price hike in December 2017, and the decrease in the number of Bal-players cause the price fall in January 2018? Or did the December 2017 price hike cause an increase in the number of In-players and the January 2018 price fall cause a decrease in the number of Bal-players? It is very interesting to ask which of these causal relationships occurred. For this purpose, it would be useful to construct a vector auto-regression (VAR) model by estimating the increasing component of player $i$, $N_i^{(\text{inc})}(t) := \sum_{j \neq i} W_{ji}(t) N_j(t)$ and the decreasing

component of player $i$, $N_i^{(\mathrm{dec})}(t) \coloneqq N_i(t)\sum_{k \neq i} W_{ik}(t)$ from Eq (18) and constructing a vector consisting of the price and the increasing and decreasing components of players. In the VAR model, Granger causality analysis can quantitatively answer whether the number of players causes price changes. These issues are briefly explained in supporting information S1 File.

## Comparison with other periods

In the present paper, we have focused on a specific target period (July 2 2017 Sunday to June 30 2018 Saturday, one year) which observed the considerable bubble and crash in the market. We performed additional analysis for different periods that cover several crypto bubbles and crashes in order to see how the market's possible changes affect the dynamics under our study. To do so, we examined the time series of the numbers of In-, Bal-, and Out players (weekly total) for Bitcoin during the following four periods, each length being one year: July 3 2016 Sunday to July 1 2017 Saturday; July 1 2018 Sunday to June 29 2019 Saturday; June 30 2019 Sunday to June 27 2020 Saturday; and June 28 2020 Sunday to June 26 2021 Saturday. The results are shown in Fig 9 for active players, and Fig 10 for regular players. We also show

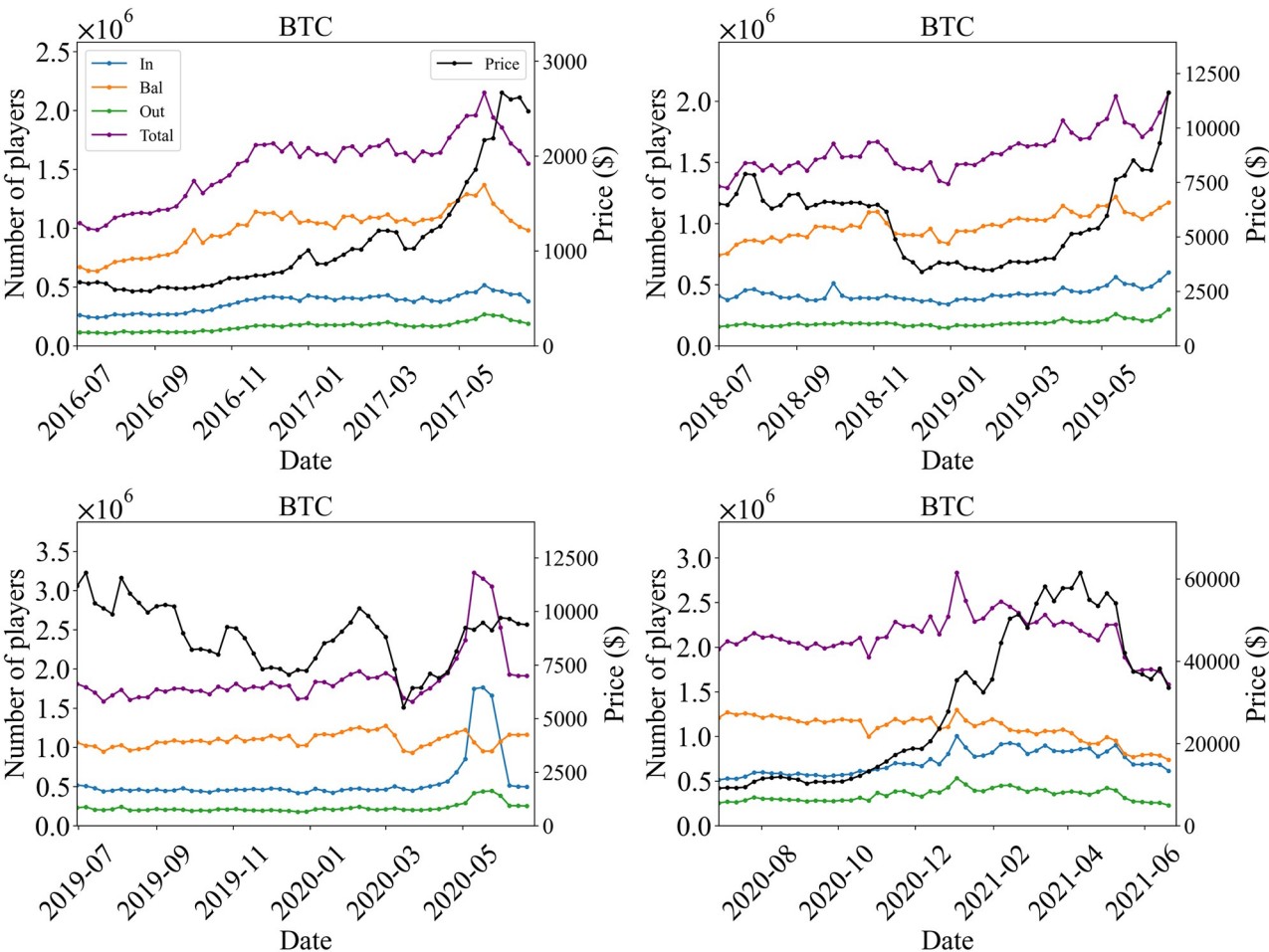

**Fig 9. Time series of the number of In-, Bal-, and Out-players (weekly total) of Bitcoin for periods (July 3 2016 Sunday to July 1 2017 Saturday), (July 1 2018 Sunday to June 29 2019 Saturday), (Jun 30 2019 Sunday to June 27 2020 Saturday), and (June 28 2020 Sunday to June 26 2021 Saturday).** The black line represents the one-week-average price.

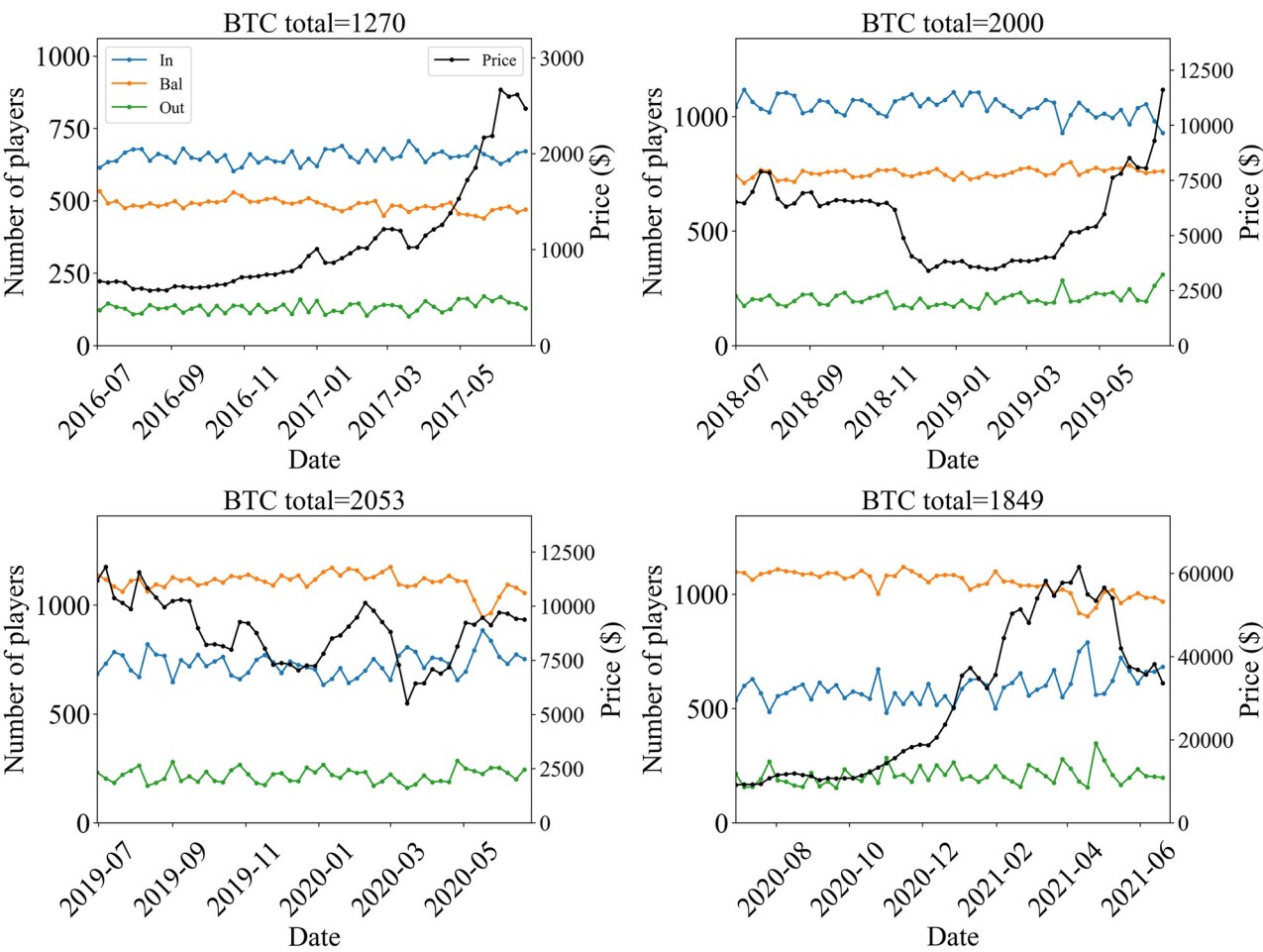

**Fig 10. Time series of the number of regular In-, Bal-, and Out-players (weekly total) of Bitcoin for periods (July 3 2016 Sunday to July 1 2017 Saturday), (July 1 2018 Sunday to June 29 2019 Saturday), (Jun 30 2019 Sunday to June 27 2020 Saturday), and (June 28 2020 Sunday to June 26 2021 Saturday).** The black line represents the one-week-average price.

correlations between the time difference of price and the time difference of the number of active players in Table 7. The period from June 30 2019 Sunday to June 27 2020 Saturday contains the epidemic of COVID-19 and presumably its shock to the economy. The price shock can be actually observed around February 2020, and an anomalous increase and decrease of In players can be seen around May 2020 as shown in Fig 9. During the period from June 27 2020 Sunday to June 26 2021 Saturday, the huge bubble and crash occurred, where prices increased by about six times and then dropped by more than 30 percent. For the regular players, the number of Bal players decreased during this bubble period. This result is similar to the one for the target period (July 2 2017 to June 30 2018). The correlation between price and the total number of players had a tendency to increase during the bubble period, as shown in Table 7 A difference is such that the correlation for $k = -1$ is not strong compared with the target period (July 2 2017 to June 30 2018).

These additional results show that the dynamics depends on the periods over a dramatic historical change of crypto (see [20] for a recent review). We would like to emphasize that our method of flow-weighted frequency and the finding of three branches can shed lights on how

**Table 7. Correlation between the time difference of price, $\Delta p(t)$, and the time difference of the number of Total, In, Bal, and Out active players, $\Delta N_i(t)$ for each target period.** $k$ represents the lag time. For example, $k = 1$ means the correlation between $\Delta p(t − 1)$ and $\Delta N_i(t)$.

| | 2016–07–03 to 2017–07–01 | | | | 2018–07–01 to 2019–06–29 | | | |
|---|---|---|---|---|---|---|---|---|
| | Total | In | Bal | Out | Total | In | Bal | Out |
| $k = 0$ | 0.23 | 0.35 | 0.08 | 0.49 | 0.52 | 0.47 | 0.34 | 0.67 |
| $k = −1$ | −0.12 | −0.14 | −0.15 | 0.14 | 0.26 | 0.10 | 0.29 | 0.24 |
| $k = 1$ | −0.20 | −0.09 | −0.22 | −0.12 | −0.12 | 0.05 | −0.24 | 0.02 |
| | 2019–06–30 to 2020–06–27 | | | | 2020–06–28 to 2021–06–26 | | | |
| | Total | In | Bal | Out | Total | In | Bal | Out |
| $k = 0$ | 0.17 | 0.01 | 0.35 | 0.24 | 0.55 | 0.64 | 0.27 | 0.59 |
| $k = −1$ | 0.18 | 0.09 | 0.21 | 0.16 | 0.26 | 0.13 | 0.22 | 0.39 |
| $k = 1$ | 0.24 | 0.19 | 0.16 | 0.04 | 0.04 | 0.21 | −0.08 | −0.06 |

the players, especially regular players, behave under dramatically varying situations of bull or bear markets of cryptoassets.

## Conclusion

In this paper, we studied the relationship between two important aspects of the cryptoasset, one in the bubble/crash of price and the other in the daily network of transactions, by using the two dominant cryptoassets of Bitcoin and XRP. The network comprises of players as nodes and flows as edges. While the network is quite huge in terms of the number of players, we focus on "regular players" who frequently appear on a weekly basis during a period of one year including bubble and crash of the price that took place during December 2017 for Bitcoin and January 2018 for XRP. We quantified each player's role with respect to outgoing and incoming flows by defining flow-weighted frequency or F-frequency. By using this measure of F-frequency, we discovered the structure of three groups of players in the diagram of flow-weighted frequency as a fact common to both of the cases of Bitcoin and XRP. In the case of Bitcoin, we found a regime switching, that is, the temporal transition from Bal-branch and In-branch was significant. By examining the identity and business activity of some regular players in the case of Bitcoin, we can observe different roles of them, namely the players balancing surplus and deficit of cryptoasset (Bal-branch), those accumulating the cryptoasset (In-branch), and those reducing it (Out-branch). Using this information, we found that the regime switching was presumably brought about by the regular players who are not necessarily dominant and stable in the case of Bitcoin, while such players are simply absent in the case of XRP. We also discuss how one can understand the temporal transitions among the three branches in a framework of VAR model, which remains an interesting future study.

## Supporting information

**S1 File. Granger causality analysis identity of Bitcoin users.**
(PDF)

## Acknowledgments

H.A. would also like to thank Hiro Inoue for technical support on data-handling. Y.F. would like to thank Aleš Janda for providing an API of `WalletExplorer.com`.

## Author Contributions

**Conceptualization:** Hideaki Aoyama.

**Data curation:** Hideaki Aoyama, Yoshi Fujiwara, Yoshimasa Hidaka, Yuichi Ikeda.

**Funding acquisition:** Yoshi Fujiwara, Yuichi Ikeda.

**Investigation:** Hideaki Aoyama, Yoshi Fujiwara, Yoshimasa Hidaka, Yuichi Ikeda.

**Methodology:** Hideaki Aoyama, Yoshi Fujiwara, Yoshimasa Hidaka, Yuichi Ikeda.

**Visualization:** Yoshimasa Hidaka.

**Writing – original draft:** Hideaki Aoyama, Yoshi Fujiwara, Yoshimasa Hidaka, Yuichi Ikeda.

**Writing – review & editing:** Hideaki Aoyama, Yoshi Fujiwara, Yoshimasa Hidaka, Yuichi Ikeda.

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
