## [Decision Letter · Decision Letter 0]

7 Jun 2022

PONE-D-22-11926Cryptoasset networks: Flows and regular players in Bitcoin and XRPPLOS ONE

Dear Dr. Fujiwara,

Thank you for submitting your manuscript to PLOS ONE. After careful consideration, we feel that it has merit but does not fully meet PLOS ONE’s publication criteria as it currently stands. Therefore, we invite you to submit a revised version of the manuscript that addresses the points raised during the review process.

Please make all data underlying the findings in your manuscript fully available.

We look forward to receiving your revised manuscript.

Kind regards,

Vygintas Gontis, Ph.D.

Academic Editor

PLOS ONE

Journal Requirements:

"The authors would like to acknowledge Ripple, which is providing financial and technical support through its University Blockchain Research Initiative. H.A. would also like to thank Hiro Inoue for technical support on data-handling. Y.F. would like to thank Aleˇs Janda for providing an API of WalletExplorer.com. The work is partially supported by JSPS KAKENHI Grant Numbers, 19K22032 and 20H02391, and by the Nomura Foundation (Grants for Social Science)."

"Y.I.: A grant "University Blockchain Research Initiative" provided by Ripple, Inc. to Kyoto University

Y.F.: JSPS KAKENHI Grant Numbers, 19K22032 and 20H02391, and the Nomura Foundation (Grants for Social Science)

All the funders had no role in study design, data collection and analysis, decision to publish, or preparation of the manuscript."

Reviewers' comments:

Reviewer's Responses to Questions

**Comments to the Author**

1. Is the manuscript technically sound, and do the data support the conclusions?

Reviewer #1: Partly

Reviewer #2: No

2. Has the statistical analysis been performed appropriately and rigorously? 

Reviewer #1: No

Reviewer #2: Yes

3. Have the authors made all data underlying the findings in their manuscript fully available?

Reviewer #1: No

Reviewer #2: No

4. Is the manuscript presented in an intelligible fashion and written in standard English?

Reviewer #1: Yes

Reviewer #2: Yes

5. Review Comments to the Author

Reviewer #1: In this paper, the authors propose an approach that uses “flow-weighted frequency” index to study the correlation between network characteristics and price of Bitcoin and XRP. In this approach, Bal-, In-, and Out-branches are found, and the temporal transitions among the three branches exist in Bitcoin, while does not exist in XRP.

Overall, this paper is well organized and contains some publishable material. However, two major problems exist. (1) Selecting the most significant period of one year as an observation period, it is doubtful whether the conclusions drawn are universal and representative. (2) the mathematical formula (11)-(13) are wrong and cannot achieve the purpose of dividing the set into three regions of 30 degrees. (3) in the Results section, the variation between some weeks and the rest of the year is not distinct enough to draw a clear conclusion.

In addition, apart from the fact that the research objects are Bitcoin and XRP, it is hard to judge whether the differences between the research methods in this paper and existing methods are large enough to be publishable.

Reviewer #2: The article covers an interesting topic of the flow of different users of the Bitcoin and Ripple networks. The discussion in this article is interesting. The authors try to connect changes in the number of different network users with the 2017/2018 market bubble. The analysis is quite attractive, but it covers only one particular example. The cryptocurrency market has changed and matured significantly since then - see for example https://doi.org/10.1016/j.physrep.2020.10.005, which makes this analysis historical. It will be really nice to see such kind of analysis for 2021 crypto bubble.

1. To have a direct comparison of price changes with changes in the number of network users, price changes could be also added in Figs. 3 and 4.

2. Did you try to calculate correlations between price changes or lagged price changes and ratio in/out in/bal out/bal players?

3. Is it possible to make any price prediction based on the number of active players or rations between various players?

4. "The ratio of Bal-players is larger before the peak, but after the peak, the ratio of In-players is larger." It is quite curious that there were more players after the peak and the price was sharply dropping?

5. "This fact suggests an abrupt change in the number of Players at the price peak seen in the upper panel of Fig. 4 for BTC regular players are due to a change in the number of non-identified regular players" this may be the effect of the influx of new players encouraged by the bull market?

Minor issues:

Line 203 P^XRP=32 not P^BTC=32

6. PLOS authors have the option to publish the peer review history of their article (what does this mean?). If published, this will include your full peer review and any attached files.

Reviewer #1: No

Reviewer #2: No

---

## [Author Response · Author response to Decision Letter 0]

26 Jul 2022

We thank the reviewer for the encouraging and thoughtful remarks and helpful suggestions. We have modified our manuscript in accordance to all of the raised points. Please find "response.pdf" for details.

Also please refer to the uploaded "Revised Manuscript with Track Changes" for the marked-up copy of our manuscript that highlights changes (in red) to the initial version, and also to the uploaded "Manuscript" for unmarked version of our revised paper without tracked changes.

---

## [Editor Report · Decision Letter 1]

3 Aug 2022

Cryptoasset networks: Flows and regular players in Bitcoin and XRP

PONE-D-22-11926R1

Dear Dr. Fujiwara,

We’re pleased to inform you that your manuscript has been judged scientifically suitable for publication and will be formally accepted for publication once it meets all outstanding technical requirements.

Kind regards,

Vygintas Gontis, Ph.D.

Academic Editor

PLOS ONE
---

## [Editor Report · Acceptance letter]

10 Aug 2022

PONE-D-22-11926R1 

Cryptoasset networks: Flows and regular players in Bitcoin and XRP 

Dear Dr. Fujiwara:

I'm pleased to inform you that your manuscript has been deemed suitable for publication in PLOS ONE. Congratulations! Your manuscript is now with our production department. 

Kind regards, 

on behalf of

Dr. Vygintas Gontis 

Academic Editor

PLOS ONE